# Efficient unified architecture for post-quantum cryptography: combining Dilithium and Kyber

Patrik Dobias, Lukas Malina and Jan Hajny

Department of Telecommunications, Brno University of Technology, Brno, Czech Republic



## ABSTRACT

As the ongoing standardization process of post-quantum schemes yields initial outcomes, it becomes increasingly important to not only optimize standalone implementations but also explore the potential of combining multiple schemes into a single, unified architecture. In this article, we investigate the combination of two National Institute of Standards and Technology (NIST)-selected schemes: the Dilithium digital signature scheme and the Kyber key encapsulation mechanism. We propose a novel set of optimization techniques for a unified hardware implementation of these leading post-quantum schemes, achieving a balanced approach between area efficiency and high performance. Our design demonstrates superior resource efficiency and performance compared to previously reported unified architecture (DOI 10.1109/TCSI.2022.3219555), also achieving results that are better than, or comparable, to those of standalone implementations. The efficient and combined implementation of lattice-based digital signatures and key establishment methods can be deployed for establishing secure sessions in high-speed communication networks at servers and gateways. Moreover, the unique and compact design that requires small hardware resources can be directly used in small and cost-effective field programmable gate array (FPGA) platforms that can be used as security co-processors for embedded devices and in the Internet of Things.

## INTRODUCTION

Today, quantum computing is a rapidly emerging area that can significantly influence the cybersecurity of information and communication (ICT) systems based on common cryptographic schemes. Asymmetric schemes based on non-polynomial (NP) time complexity problems, such as the integer factorization, discrete logarithm, and elliptic curve discrete logarithm problems are vulnerable quantum computer attacks that leverage Shor's algorithm (see more in *Bernstein & Lange, 2017*). Therefore, National Institute of Standards and Technology (NIST) has recently standardized the Kyber (*Bos et al. (2018)*) and Dilithium (*Ducas et al. (2018)*) schemes as Module-Lattice-based Key-Encapsulation Mechanism (ML-KEM) in *NIST (2024b)* and Module-Lattice-based Digital Signature Algorithm (ML-DSA) in *NIST (2024a)*, respectively, along with Practical Stateless Hash-based Signatures (SPHINCS+) (*Zhang, Cui & Yu, 2022*) as Stateless Hash-Based Digital Signature Algorithm (SLH-DSA) in *NIST (2024c)*. These quantum-resistant schemes

Corresponding author
Patrik Dobias, xdobia13@vut.cz

should substitute the legacy asymmetric schemes such as Rivest–Shamir–Adleman (RSA), Digital Signature Algorithm (DSA), Elliptic Curve Digital Signature Algorithm (ECDSA) *etc*. in the next few years during the post-quantum transition period, as foreseen in reports, and recommendations (*ANSSI, 2022*; *NSA, 2022*). Deploying the quantum-resistant schemes will also be motivated by harvest now, decrypt later attacks. The results of the standardization have increased the demand for efficient and secure implementations of these cryptographic schemes in practical applications. Software implementations are popular for their flexibility and ease of deployment. However, the significant computational requirements of post-quantum cryptographic (PQC) algorithms make software-only approaches insufficient for many high-performance or resource-constrained environments. Hardware-based implementations of PQC on Field Programmable Gate Array (FPGA) platforms offer distinct advantages, particularly by providing lower latencies and, therefore, higher throughput. High-speed communication and operation can be essential for servers and gateways that manage hundreds to thousands of security sessions and have to perform key establishment and signing/verification phases efficiently. This use case is demonstrated in Fig. 1. Additionally, hardware acceleration improves security by reducing the susceptibility to certain types of side-channel attacks that are often present in software implementations. Given that ML-KEM and ML-DSA belong to the same cryptographic family, they are well suited for unified hardware architectures, which can reduce resource usage and streamline security processes by securing shared components only once.

In this work, we focus on integrating these schemes into a single hardware implementation that can be compact and efficiently (low hardware resources, low latency) provide basic security operations, *i.e.*, key establishment and data signing/verification that are usually required while establishing secure communication sessions. The work aims at two research questions (RQ): *RQ1) are there any new optimization approaches that can be designed and applied in the hardware-implementations of lattice-based schemes (Kyber and Dilithium)? RQ2) how can the lattice-based standards (Kyber with Dilithium) be efficiently combined and how efficient can this combination be on the hardware (FPGA) platforms in comparison with standalone HW implementations?*

The rest of this article is organized as follows: "Background" introduces background of lattice-based schemes and their main phases. "Optimized Unified Hardware Architecture" presents new optimization techniques that are proposed and deployed in our unified hardware architecture, and deals with RQ1 and RQ2. "Results and Comparison" shows results and compares our solution with related works, and deals with RQ2. "Discussion" discusses the practical deployment, limitations, implementation attacks and future open problems. In "Conclusion" we conclude this work and present our next steps.

### Related work

By finishing the NIST PQC standardization, finalist PQC schemes and winners have been implemented on various FPGA platforms, tested in various architectures, or co-designed for Hardware/Software (HW/SW) co-processors. Related works have investigated mostly

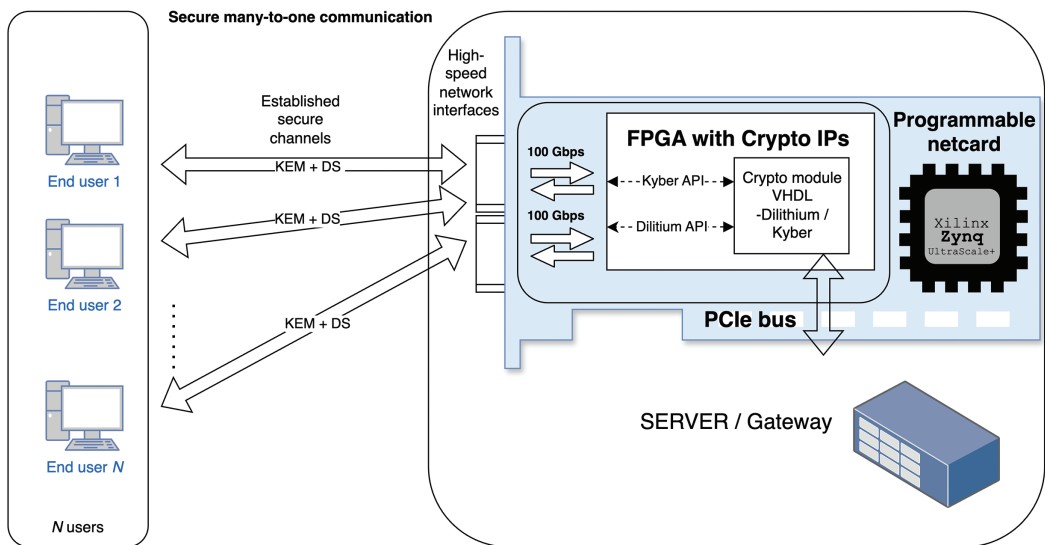

**Figure 1 Use case: server-side handling of multiple users with varying security requirements.**

implementation efficiency (*e.g.*, speed, area, latency) and security (*e.g.*, preventing side channel attacks, hardware trojans, fault injections *etc.*).

### *Standalone hardware implementations*

Most research on hardware implementations of PQC schemes has focused on designs tailored to individual schemes. Several studies have explored hardware implementations of Dilithium, such as *Ricci et al. (2021)*, *Beckwith, Nguyen & Gaj (2021)*, *Land, Sasdrich & Güneysu (2021)*, *Zhao et al. (2021)* and *Gupta et al. (2023)*, and Kyber, such as *Xing & Li (2021)*, *Guo, Li & Kong (2022)*, *Bisheh-Niasar, Azarderakhsh & Mozaffari-Kermani (2021)*, *Dang, Mohajerani & Gaj (2023)*, *Ni et al. (2023)* and *Nguyen et al. (2024)*.

For Dilithium, *Ricci et al. (2021)* presented the first dedicated hardware implementation, optimizing the number theoretic transform (NTT) but employing separate architectures for each phase, which would increase hardware costs when combined into a complete scheme. *Beckwith, Nguyen & Gaj (2021)* prioritized high performance, achieving reduced latency by splitting the rejection loop into a two-stage pipeline. *Land, Sasdrich & Güneysu (2021)* proposed a more compact design by optimizing the NTT to use more DSP blocks, thereby reducing LUT and FF usage. Similarly, *Zhao et al. (2021)* introduced a segmented pipelined processing method, dividing algorithms into multiple stages and optimizing key components such as `NTT`, `SampleInBall`, and `Decompose`. However, both (*Land, Sasdrich & Güneysu, 2021*) and *Zhao et al. (2021)* omitted coefficient packing/unpacking, which would need to be implemented separately for compatibility with software implementations. *Gupta et al. (2023)* achieved the lowest area usage to date for Dilithium by employing optimization techniques such as resource and control logic sharing, module fusion, and pre-computed LUTs.

For Kyber, *Xing & Li (2021)* presented a compact implementation that unified NTT and pointwise multiplication (PWM) operations, using only a single pair of butterfly units. *Guo, Li & Kong (2022)* enhanced this design with a resource-efficient modular reduction

and further optimized the unified NTT-PWM module. *Bisheh-Niasar, Azarderakhsh & Mozaffari-Kermani (2021)* focused on improving the NTT and proposed a hardware-friendly modular reduction approach. *Dang, Mohajerani & Gaj (2023)* extended their work to include benchmarking across multiple schemes, implementing Kyber, NTRU, and Saber. They proposed a memory-efficient NTT design by incorporating reordering units before and after the operation, thus reducing the memory requirements. *Ni et al. (2023)* achieved the highest throughput for Kyber by employing efficient pipelining, while (*Nguyen et al., 2024*) developed the most lightweight implementation by designing a compact Secure Hash Algorithm 3 (SHA3) hash function architecture and introducing the first non-memory-based iterative NTT, significantly reducing memory requirements.

A key limitation of these standalone designs is that in practical applications, both digital signature algorithms (DSAs) and key encapsulation mechanisms (KEMs) are often required. Thus, using these single-purpose implementations would require combining them into a single firmware, resulting in additional resource utilization.

### Unified hardware architectures

Several studies have focused on the unification of hardware architectures, including those by *Basso et al. (2021)*, *Aikata et al. (2023b)*, *Karl et al. (2024)*, *Aikata et al. (2023a)* and *Mandal & Roy (2024)*.

*Basso et al. (2021)* implemented a unified polynomial multiplier for the Dilithium and Saber schemes, proposing modifications to Dilithium's NTT multiplier, so that it can also be used for Saber multiplication. This approach resulted in a negligible probability of incorrect results. Building on this, *Aikata et al. (2023b)* developed a complete unified architecture for Dilithium and Saber (*D'Anvers et al. (2018)*). Although Saber was not selected for standardization by NIST, their work provides valuable insights into integrating seemingly incompatible schemes.

The unification of multiple signature schemes was explored by *Karl et al. (2024)*, where the authors unified the Dilithium signature scheme across all security levels with the Falcon verification phase for specific use cases.

In the most relevant work, *Aikata et al. (2023a)* presented the first hardware implementation of a unified architecture for Dilithium and Kyber. The authors identified key components that would significantly impact performance, such as polynomial multiplication and Keccak with rejection samplers, and optimized their sharing. Their results achieved a single implementation that supports all security levels of Dilithium and Kyber, with performance comparable to state-of-the-art standalone implementations. Building on this, *Mandal & Roy (2024)* focused on further optimizing the unified NTT multiplication for Dilithium and Kyber. They explored the trade-offs between area consumption and performance by varying the number of butterfly units, achieving lower area usage and better performance than the approach in *Aikata et al. (2023a)*.

This article enhances the optimization of the unified architecture in multiple levels in hardware implementation and so far provides the most efficient trade-off between efficiency and the amount of required HW resources.

## Contribution

Our contributions are twofold and summarized as follows:

1. We introduce novel approaches to enhance the level of resource reuse within the unified architecture. Mainly we use a new memory management, reducing the memory used, polynomial arithmetic, sample and compression units, and efficient operations schedule. These optimization steps cause a 50% reduction in BRAM usage in our hardware implementation compared to *Aikata et al. (2023a)*.

2. We present the most compact and high-performance hardware implementation of a unified architecture that supports both Dilithium and Kyber for all security levels. Our solution is compared with related works and also standalone HW implementations. Our efficient solution can also be deployed in small, cost-effective, and low-density FPGA platforms.

## BACKGROUND

Despite the standards containing Dilithium and Kyber using different names, we will retain their original designations for consistency with existing articles. This section provides a brief overview of these schemes, highlighting their similarities that are used in the unified design.

### Dilithium

Dilithium is a lattice-based digital signature scheme whose security relies on the hardness of module learning with errors (M-LWE) and shortest integer solution (SIS) problems. The scheme works in three phases: key generation, signing and verification, each involving multiple operations on polynomials. Below is a brief description of the key operations involved in these phases.

#### Polynomial arithmetic

All arithmetic operations on polynomials are performed over the ring $R_{8380417} = \mathbb{Z}_{8380417}[X]/(X^{256} + 1)$. To enable faster polynomial multiplication, the NTT is used. Since Dilithium includes a $512^{th}$ root of unity, the polynomial multiplication is carried out using the complete-NTT, where the coefficients in the NTT domain correspond to zeroth-degree polynomials, thus the multiplication process requires only 256 pointwise multiplications.

#### Coefficients sampling

Dilithium uses two forms of coefficient sampling, both of which involve rejection sampling. The first occurs during the generation of the matrix $A$, where rejection sampling is applied to the output of the `SHAKE-128` hash function. The second occurs during the generation of the secret key and error vectors, where rejection sampling is applied to the output of the `SHAKE-256` hash function. Additionally, a `SampleInBall` operation is used to sample the challenge polynomial; however, this operation differs significantly and is handled by its own dedicated unit.

### Coefficients rounding

Two types of coefficient rounding are used in Dilithium. The first, `Power2round`, breaks a coefficient $r$ into two parts: high bits $r_1$ and low bits $r_0$, such that $r = r_1 \cdot 2^d + r_0 \bmod q$. The second, `Decompose`, which similarly decomposes a coefficient into two parts, where $r = r_1 \cdot (2\gamma_2) + r_0 \bmod q$.

### Coefficients unpacking/packing

To reduce the size of keys and signatures, and consequently lower memory and bandwidth requirements, the coefficients are unpacked and packed to and from the byte arrays. During these operations, only a specific number of bits from the coefficients are used. In particular, unpacking and packing operations are required for coefficients of 20-bit, 18-bit, 13-bit, 10-bit, 6-bit, 4-bit, and 3-bit lengths.

## Kyber

Kyber is a lattice-based key encapsulation scheme whose security relies on the hardness of the M-LWE problem. The scheme works in three phases: key generation, key encapsulation and key decapsulation, each involving multiple operations on polynomials. Below is a brief description of the key operations involved in these phases.

### Polynomial arithmetic

All arithmetic operations on polynomials are performed over the ring $R_{3329} = \mathbb{Z}_{3329}[X]/(X^{256} + 1)$. Similarly to Dilithium, NTT is used to enable faster polynomial multiplication. However, since Kyber does not include a $512^{th}$ root of unity, the polynomial multiplication is carried out using the incomplete-NTT, where the coefficients in the NTT domain correspond to first-degree polynomials, thus the multiplication process is more complex.

### Coefficients sampling

Similarly to Dilithium, Kyber also uses two types of coefficient sampling, but only one uses rejection sampling. The first occurs during the generation of the matrix $A$, where rejection sampling is applied to the output of the `SHAKE-128` hash function. The second is used for sampling the secret key and error vectors, during which only a specific number of bits (without rejection) are sampled from the output of `SHAKE-256`.

### Coefficients compression/decompression

Kyber uses compression to reduce the coefficient sizes by discarding the least significant bits. The compression function is defined as $\lceil (2^d/q) \cdot x \rceil \bmod 2^d$, and decompression as $\lceil (q/2^d) \cdot x \rceil \bmod 2^d$. Decompressing a coefficient and then compressing it again always gives the same value.

### Coefficients decoding/encoding

To reduce the size of keys and ciphertexts, coefficients are decoded and encoded to byte arrays. During these operations, only a specific number of bits from the coefficients are used. The decoding and encoding process must support coefficients of 12-bit, 11-bit, 10-bit, 5-bit, 4-bit, and 1-bit lengths.

### Changes in NIST standards

There were few changes to the standards of ML-KEM and ML-DSA compared to the original proposals of Kyber and Dilithium.

For ML-KEM, the Fujisaki-Okamoto (FO) transformation has been modified with the removal of additional message hashing. Furthermore, input validation checks were introduced, although our implementation omits these, as it is specified that such checks do not need to be performed directly by the party. Instead, we assume that these correctness checks would be handled by the controller component within our design.

In the case of ML-DSA, there have been parameter size adjustments, with the primary modification being the support for both deterministic and non-deterministic variations of the algorithm.

## OPTIMIZED UNIFIED HARDWARE ARCHITECTURE

In this section, our unified hardware architecture is proposed and described. It is based on the work of *Aikata et al. (2023a)*, but we employ new optimization techniques to make the design more compact with a better performance.

Concretely, we have improved and optimized these five parts: memory management, polynomial arithmetic unit, polynomial sample unit, compression unit, and operation schedule. In addition to those major improvements, we have made further minor improvements. The following subsections describe major and minor optimization steps. Figure 2 presents a top-level overview of the architecture, highlighting the unified units shared by Dilithium and Kyber, Dilithium-specific units such as `SampleInBall`, and the memories used to store polynomials.

### Memory management

Our design employs two dual-port RAMs for coefficient storage: a main memory and a temporary memory, both with a width of 96 bits. This configuration allows each RAM to store either 4 Dilithium coefficients or 8 Kyber coefficients, matching the number of coefficients processed in parallel by other units. The main memory has a depth of 4,096, sufficient to store all necessary polynomials for any phase of Dilithium or Kyber operations. In contrast, the temporary memory, with a depth of 512, is used to store temporarily polynomials, that are decoded from input or sampled. This allows the polynomial arithmetic unit to process polynomials stored in the main memory in parallel with the preparation of new polynomials in the temporary memory. This arrangement optimizes the operation schedule by enabling continuous utilization of the arithmetic unit, thereby reducing idle times that would otherwise occur if the unit had to wait for polynomial loading or sampling. The selected depth of the temporary memory ensures that a sufficient number of polynomials is available in the memory to allow this continuous processing.

As Dilithium requires significantly more memory to store the polynomials, the memory optimizations were done only for Dilithium phases. On the other hand, during Kyber phases, the additional space is used to enable efficient processing.

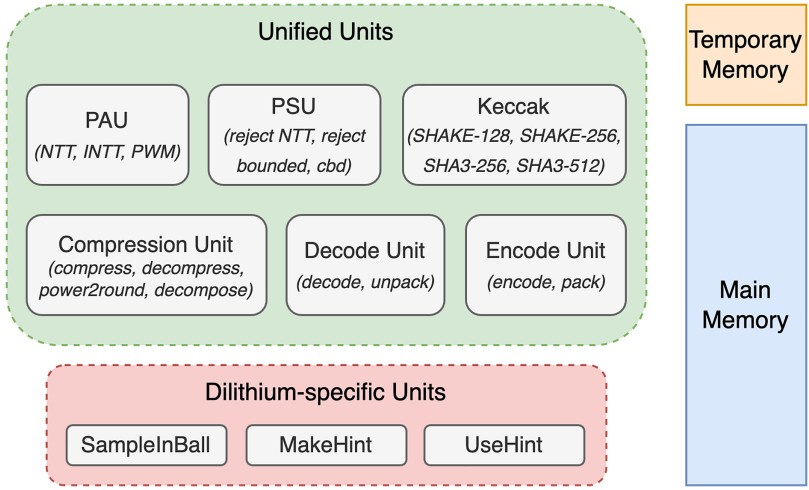

**Figure 2** Overview of top-level units, illustrating both unified and dilithium-specific units, along with memory details.

## Polynomial arithmetic unit

The polynomial arithmetic unit (PAU) performs polynomial multiplication with optional in-place polynomial addition or subtraction. The design of this unit combines the proposals of *Aikata et al. (2023a)* and *Mandal & Roy (2024)* to allow the unified processing of Dilithium and Kyber polynomials, while the integration coefficients reorder the units from *Dang, Mohajerani & Gaj (2023)* to minimize the use of RAM. The unit uses two unified butterfly units, which enable processing of four Dilithium or 8 Kyber coefficients every clock cycle.

### Unified butterfly unit

The unified butterfly unit (BFU) can perform a butterfly operation, that is a fundamental operation in the NTT, on either two Dilithium coefficients or four Kyber coefficients. It uses the modular multiplier and reduction proposed by *Aikata et al. (2023a)* and the improved modular adder/subtractor from *Mandal & Roy (2024)* that fully utilizes fast carry chains. The architecture of this unit is illustrated in Fig. 3. For clarity, control signals that determine whether Dilithium or Kyber coefficients are being processed are omitted from the figure, but are integral to the design. These signals are required for all of the operations to determine the correct modulus and specify whether operations are performed on single 23-bit coefficients (for Dilithium) or two 12-bit coefficients (for Kyber). This adaptability is a key feature allowing the BFU to handle both schemes without requiring separate hardware blocks. After modular reduction, additional coefficients can be added or subtracted, enabling in-place operations during polynomial multiplication.

To remove the butterfly feedback unit described in *Aikata et al. (2023a)* and thus reduce the area of the PAU, Kyber multiplication is performed in two iterations, with intermediate results stored in memory. Note that no additional memory is needed for these intermediate results, as the memory requirements for Dilithium are substantially larger, as outlined in

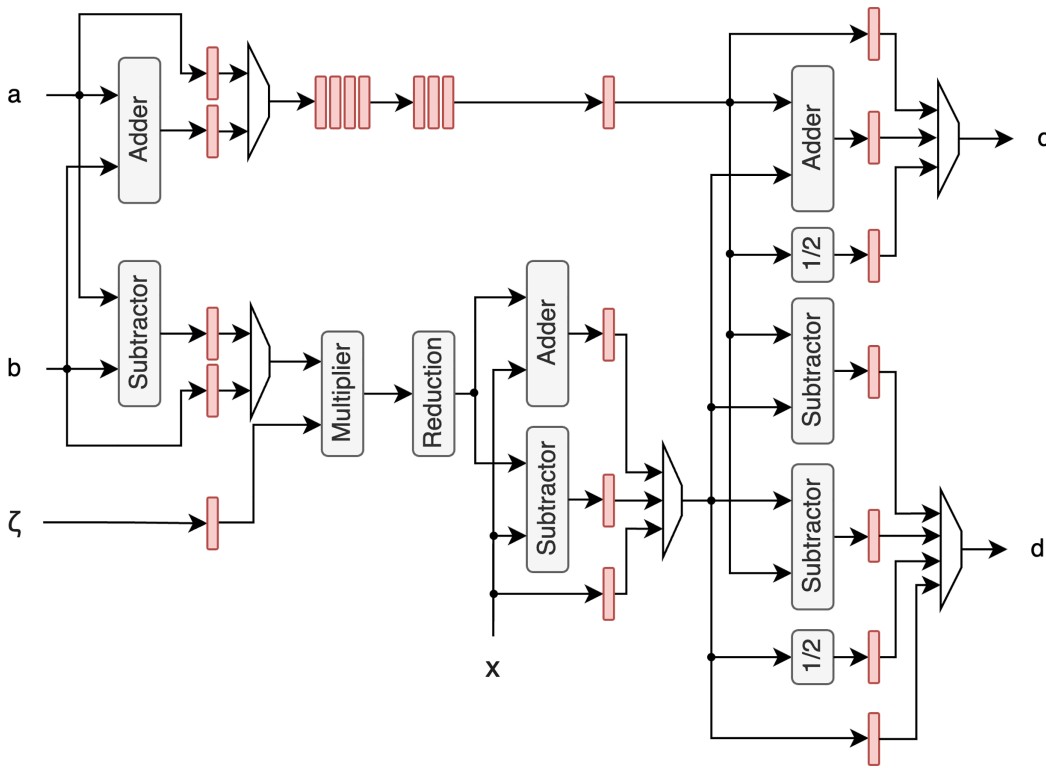

**Figure 3 Butterfly unit architecture.** Registers between stages are highlighted with red color.

the previous section. This allows the intermediate results for Kyber to be stored in memory regions that remain unused during Kyber's computation.

### Optimized memory mapping

Both unified polynomial arithmetic units proposed in *Aikata et al. (2023a)* and *Mandal & Roy (2024)* use two dual-port RAMs. While *Aikata et al. (2023a)* do not specify how they achieve fully pipelined memory access, *Mandal & Roy (2024)* provide an efficient memory mapping scheme, which changes the order of coefficients in both RAMs. However, our design allows one to write to only one RAM, making their approach inapplicable. To enable fully pipelined memory access for NTT/INTT, we utilize the head and tail reorder method from *Dang, Mohajerani & Gaj (2023)*, extending them to handle coefficient access during multiplication. The key concept of this reordering strategy is to use a 96-bit register to temporarily store unprocessed coefficients. This ensures that coefficients are stored consistently in the same order throughout the whole computation process.

An example of the coefficients' flow during the first iteration of Dilithium's NTT is given in Fig. 4. The figure presents two pipelines: the input pipeline, which shows how the coefficients are reordered before processing by the BFUs, and the output pipeline, which shows how the coefficients are reordered afterwards. It can be observed that the input and output order of the coefficients is preserved.

**Figure 4 Coefficients flow during first iteration of Dilithium's NTT.** Red color highlights coefficients that are stored in registers.

## Polynomial sample unit

The polynomial sample unit (PSU) handles all polynomial samplings required for both Dilithium and Kyber, except for the `SampleInBall` function, which has a significantly different implementation. Although the PSU's performance is not critical, it must sample coefficients faster than the PAU processes them. This is achievable through efficient scheduling of operations, which will be discussed in a later section, and the use of temporary memory, as previously mentioned.

The sampling process in the PSU requires outputs from SHAKE-128 and SHAKE-256 hash functions. To implement these and other necessary hashing functions, such as SHA3-256 and SHA3-512, we used our previous design from *Ricci et al. (2021)*, with slight modifications. Given that sampling performance is not a critical factor, we chose to use the book-keeping approach from *Aikata et al. (2023b)*, reading data in 64-bit transactions. This choice reduces combinatorial logic without negatively impacting the overall system, as reduced throughput affects only the first polynomial sampled, a negligible factor throughout the sampling process.

The architecture of this unit is shown in Fig. 5. The *bytes* signal, representing 8 bytes of hash output, is processed in parallel by the CBD, `Reject Bounded`, and `Reject NTT` sampling units. Each of these units uses a bit from the *mode* signal to determine which parameters to apply, as each sampling method can operate in one of two modes: CBD can be set to $\eta = 2$ or $\eta = 3$, `Reject Bounded` can be set to $\eta = 2$ or $\eta = 4$, and `Reject NTT` can use rejection bounds $q = 3{,}329$ for Kyber or $q = 8380147$ for Dilithium. Afterward, the output from the corresponding unit is selected based on the *mode* signal and is then output from the full unit.

Furthermore, we decided not to integrate Keccak with coefficient sampling, diverging from the approach in *Aikata et al. (2023a)*. By keeping Keccak separate, it can serve multiple hashing purposes, thereby enhancing the versatility of the designed cryptographic core.

## Compression unit

The compression unit unifies all operations that modify the bit size of the coefficients, be it by reducing or extending it. It performs compression and decompression of all coefficient sizes for Kyber, and `Decompose`, `Power2round` and coefficient modification before

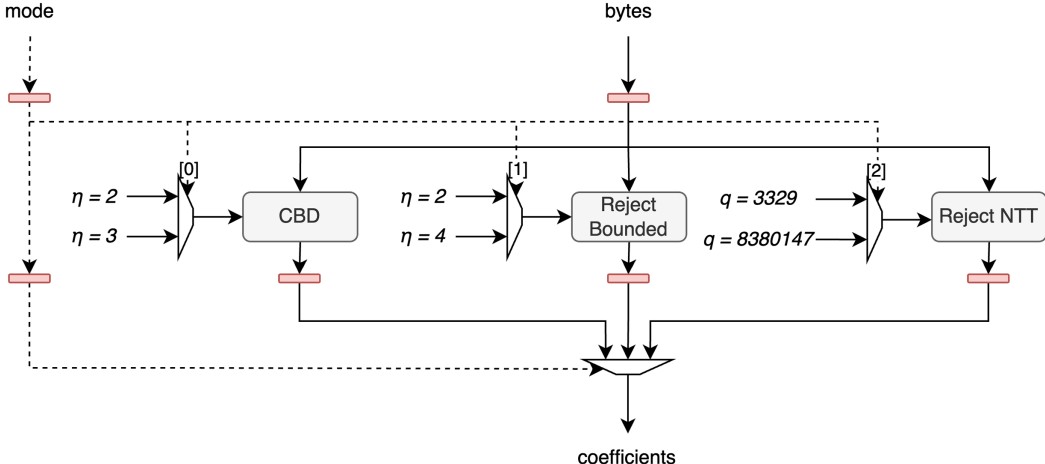

**Figure 5 Architecture of polynomial sample unit.**               

packing or after unpacking in Dilithium. This combination results in a minor reduction in area utilization, but more importantly, it simplifies the architecture by reducing the complexity when switching between components.

For the compression of the Kyber coefficient, we use sufficient precision to convert a division by $q$ into multiplication and shift operations. To further reduce complexity, we propose a new optimized algorithm, shown in Algorithm 1, which takes advantage of the lower precision required for different bit sizes. Unlike *Aikata et al. (2023a)*, our approach uses a single constant for multiplication, regardless of the compression level. This reduces both the complexity and critical paths in the compression process.

## Efficient operations schedule

To maximize the utilization of components and minimize delays between operations, the order of operations was carefully scheduled. The temporary memory, discussed earlier, plays a crucial role in enabling parallel loading/sampling of polynomial coefficients while previously loaded/sampled ones are being processed. This allowed us to schedule operations so that the PAU runs continuously after the first polynomial is loaded.

Two examples of efficient scheduling are shown in Fig. 6, which both illustrate vector-matrix multiplication during the key generation phase, one for the Dilithium scheme and the other for the Kyber scheme. The rectangles represent the polynomials sampled or processed during arithmetic operations, with the order of operations shown sequentially over time. Two memory units are depicted as lines, with arrows indicating storing or reading from these memories. In both examples, it can be observed that after sampling the initial polynomial $s_0$, the PAU starts processing and runs continuously as the subsequent polynomials are sampled in the meantime. In this first example, two key points are important:

1. The vector and matrix polynomials are sampled interleaved because the sampling of the matrix polynomial takes slightly more cycles than the multiplication. This can be compensated for by the depth of the temporary memory. However, if all the

**Algorithm 1** **Optimized compression algorithm.**

**In:** $x \in \mathbb{Z}_{3329}$, $d \in \{1, 4, 5, 10, 11\}$

**Out:** $y = \lceil (2^d/3329) \cdot x \rceil$

1: $t = x + (x \ll 2) + (x \ll 3)$

2: $t = (t \gg 7) + (t \gg 6) + (t \gg 3) + (t \ll 1) + (t \ll 4) + (x \ll 10) + (1 \ll (21 - d))$

3: $y = t \gg (22 - d)$

4: **return** $y \quad 2^d$

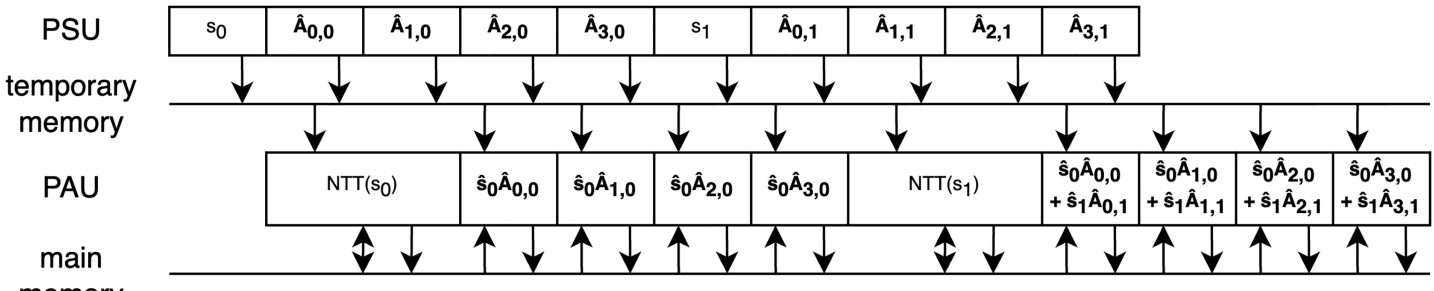

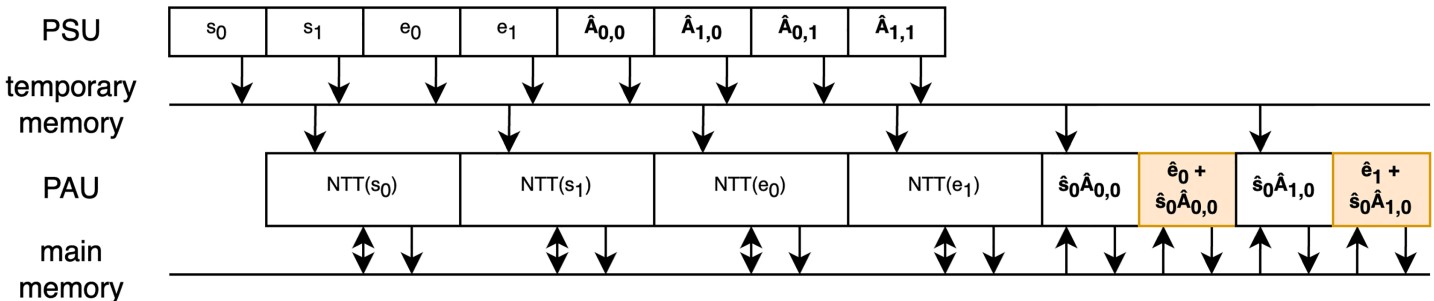

**Figure 6 Operations schedule of vector-matrix multiplication during Dilithium and Kyber key generations.**

multiplications happened without interleaving with the NTT operation as it was done in previous work, the depth of the temporary memory would have to increase.

2. During the second phase of multiplication, in-place addition is performed by the butterfly units. This is possible because, while coefficients are loaded from temporary memory, the coefficients in the main memory can be accessed simultaneously.

The second example focuses on the Kyber scheme. Since polynomial multiplication in Kyber occurs in two phases (with the second phase highlighted in orange), it takes more cycles than rejection sampling, making interleaving unnecessary. Additionally, to take advantage of in-place addition with the error vector, it is beneficial to first sample and

convert the vectors to the NTT domain before performing the multiplications, as interleaving this part would unnecessarily complicate the scheduling. Similarly to the first example, in-place addition is used to enable efficient processing.

### Minor optimizations

Some minor optimizations were further applied to reduce the total area consumption as well as remove critical paths to increase the design working frequency.

#### Data storing

As all of the internal data are processed in 64-bit transactions, we were able to use a 64-bit wide Look-Up Table Random Access Memory (LUTRAM) to store all the necessary data (seeds for key generations and for samplings, challenge in Dilithium,…) removing the need for 2,048 registers that would be needed otherwise.

#### Encode and decode units

For the implementation of encode and decode units, we followed the approach proposed by *Aikata et al. (2023a)*, with a slight modification. In our design, we combine these units with the pack and unpack units, respectively. Similarly as with the compression unit, this combination results in a minor reduction in area utilization, but more importantly, it simplifies the architecture by reducing the complexity when switching between components.

#### Critical path removal

Critical paths in the design were identified through multiple iterations using Vivado timing reports. Registers were inserted into these paths during the implementation process to incrementally improve the overall frequency of the design.

## RESULTS AND COMPARISON

In this section, we present results of our unified architecture. Our design was implemented using Very High-Speed Integrated Circuit Hardware Description Language (VHDL) and tested using the reference C implementations, which were modified to match the changes in NIST standards, alongside the Python-based cocotb framework. The results are obtained using Vivado 2022.2 after completing the place-and-route phase of the implementation. For a fair comparison with the original unified architecture for Dilithium and Kyber proposed by *Aikata et al. (2023a)*, the same target platform, UltraScale+ ZCU102, was used. Furthermore, to demonstrate the scalability and usability of our design in highly resource-constrained environments, we also targeted the Artix-7 (XC7A35T) platform, enabling direct comparisons with compact-focused designs.

### Results

Table 1 presents the hardware resource utilization of the primary components in our design, as well as the total resource usage for the top-level component that implements both the Dilithium and Kyber schemes. It is important to note that the sum of the

**Table 1 Hardware resources utilization of designed components.**

| Component | LUT | FF | DSP | BRAM |
|---|---|---|---|---|
| Compress | 2,418 | 1,061 | 0 | 0 |
| Decode | 583 | 284 | 0 | 0 |
| Encode | 786 | 380 | 0 | 0 |
| Keccak | 4,135 | 1,713 | 0 | 0 |
| MakeHint | 784 | 286 | 0 | 0 |
| Polynomial arithmetic | 2,856 | 1,347 | 4 | 0 |
| Sample | 864 | 456 | 0 | 0 |
| SampleInBall | 177 | 87 | 0 | 0 |
| UseHint | 476 | 233 | 0 | 0 |
| ValidityCheck | 139 | 129 | 0 | 0 |
| **Top** | **17,138** | **6,559** | **4** | **12.5** |

individual components' resource usage is lower than the total design utilization, as some utility components, such as memories and state machines, are not included in the individual breakdown. The top-level component utilizes 17,138 Look-Up Table (LUTs), 6,559 Flip-Flop (FFs), 4 Digital Signal Processing (DSPs), and 12.5 Block Random Access Memories (BRAMSs), with the largest impact from the Keccak, polynomial arithmetic, and compression units. While the Keccak and polynomial arithmetic units have the most significant impact in previous research and are therefore the focus of most optimizations, the impact of the compression unit in our design is notable because it combines multiple operations. These include Kyber's compression/decompression and Dilithium's power2round and decompose, which are typically reported as separate operations in existing studies. For the Artix-7 platform, this represents 82% LUT, 16% FF, 5% DSP and 25% BRAM utilizations. Fully routed design is shown in Fig. 7 with the top three highest utilization components highlighted.

Table 2 presents the performance results for all security levels of Kyber and Dilithium with our implementation targeting a working frequency of 375 MHz. The table shows both the number of cycles and the corresponding execution time for each phase. For the Dilithium signing phase, we report the best-case scenario, where the signature is valid in a single iteration.

## Comparison

When comparing our results with existing work, we primarily compare them with designs of unified architectures. However, only the works of *Aikata et al. (2023a)* and *Aikata et al. (2023b)* implemented hardware designs that can be compared with ours that we know of. However, it is important to note that the *Aikata et al. (2023b)* design unifies Dilithium with Saber, making a direct comparison not entirely fair. Therefore, we also include comparisons with Dilithium-only and Kyber-only implementations for a broader evaluation.
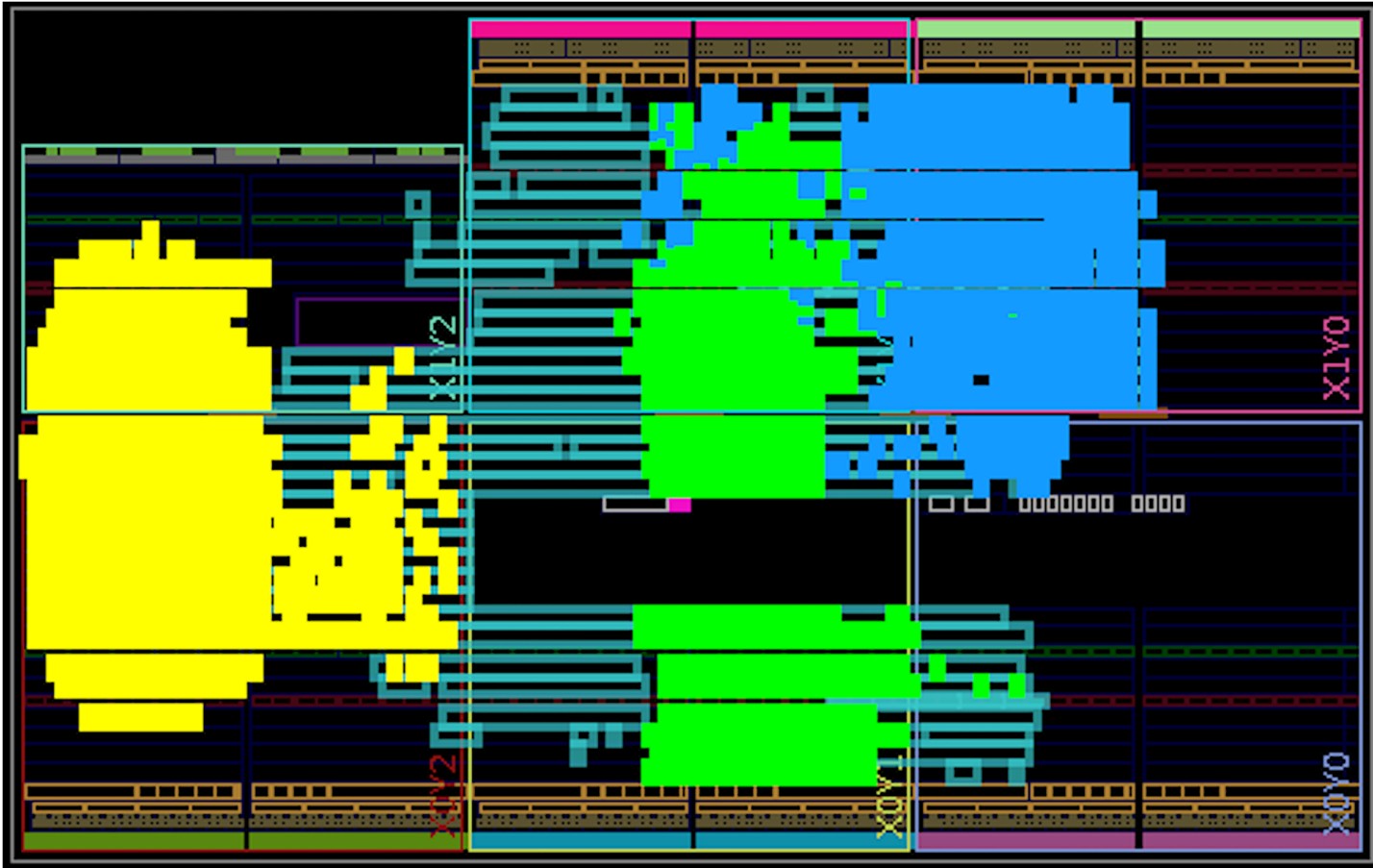

**Figure 7 Components placement after the place-and-route step for the Artix-7 board.** The three highest utilization components are highlighted: compress (green), Keccak (yellow), and polynomial arithmetic (blue).

**Table 2 Performance results of Dilithium and Kyber.**

| Variant | Key generation | | Sign/Encapsulation | | Verify/Decapsulation | |
|---|---|---|---|---|---|---|
| | Cycles | μs | Cycles | μs | Cycles | μs |
| Dilithium-2 | 9,279 | 24.7 | 22,508 | 60.0 | 10,137 | 27.0 |
| Dilithium-3 | 15,646 | 41.7 | 33,169 | 88.5 | 16,728 | 44.6 |
| Dilithium-5 | 24,278 | 64.7 | 49,124 | 131.0 | 26,986 | 72.0 |
| Kyber-512 | 2,146 | 5.7 | 2,638 | 7.0 | 3,740 | 10.0 |
| Kyber-768 | 3,523 | 9.4 | 4,087 | 10.9 | 5,544 | 14.8 |
| Kyber-1024 | 5,195 | 13.9 | 5,831 | 15.5 | 7,644 | 20.4 |

## *Comparison with unified hardware implementations*

When comparing our design with the combined architecture of *Aikata et al. (2023a)*, our implementation outperforms theirs in both area and performance for both Dilithium and Kyber. Specifically, our memory management strategy reduces BRAM usage by nearly half. In addition, efficient operation scheduling reduces the clock cycles required for

**Table 3 Comparison of unified hardware implementations.**

| Ref | Device | Area (LUT/FF/DSP/BRAM) | Frequency[MHz] | Variant | Performance $\mu s$ |
|---|---|---|---|---|---|
| **TW** | ZUS+ | 17.1k/6.6k/4/12.5 | 375 | Dilithium-2 | 24.7/60.0/27.0 |
| | | | | Dilithium-3 | 41.7/88.5/44.6 |
| | | | | Dilithium-5 | 64.7/131.0/72.0 |
| | | | | Kyber-512 | 5.7/7.0/10.0 |
| | | | | Kyber-768 | 9.4/10.9/14.8 |
| | | | | Kyber-1024 | 13.9/15.5/20.4 |
| *Aikata et al. (2023b)* | ZUS+ | 18.5k/9.3k/4/24 | 200 | Dilithium-2 | 70.9/151.6/75.2 |
| | | | | Dilithium-3 | 114.7/237.0/127.6 |
| | | | | Dilithium-5 | 194.2/342.1/228.9 |
| | | | | LightSaber | 29.6/40.4/58.3 |
| | | | | Saber | 54.9/69.7/94.9 |
| | | | | FireSaber | 87.6/108.0/139.4 |
| *Aikata et al. (2023a)* | ZUS+ | 23.3k/9.8 k/4/24 | 270 | Dilithium-2 | 54.1/117.2/57.1 |
| | | | | Dilithium-3 | 87.5/180.5/96.8 |
| | | | | Dilithium-5 | 147.2/258.3/172.9 |
| | | | | Kyber-512 | 12.6/18.4/25.2 |
| | | | | Kyber-768 | 23.2/29.1/41.8 |
| | | | | Kyber-1024 | 33.7/42.0/51.5 |

computation, while critical path removal and pipelining maximize operating frequency, resulting in substantial performance improvements. In particular, for the highest security level, our implementation achieves a performance improvement of 2.2x/1.9x/2.4x for Dilithium across all phases and 2.3x/2.8x/2.5x for Kyber across all phases, along with a reduction in area usage by 27% in terms of LUTs and 33% in terms of FFs. In the case of *Aikata et al. (2023b)*, we compare only the Dilithium portion of their design. Our implementation again achieves superior resource utilization and performance, with approximately half the BRAM usage and even more significant gains in performance. Although differences in area utilization are less pronounced, our design still exhibits better results. In Table 3, both implementations are compared with our work.

### Comparison with hardware implementations of dilithium

In Table 4, we detail the comparison with Dilithium standalone hardware implementations. *Beckwith, Nguyen & Gaj (2021)* split the signature generation process into two stages, achieving a higher performance than our design at the expense of significantly higher area utilization. Another high-performance design by *Li et al. (2024)* uses parallel instruction execution to minimize latency, although it requires nearly double the LUT usage and six times more DSP blocks than our implementation. Moreover, both of these designs operate at lower frequencies.

Lastly, *Gupta et al. (2023)* offers a design with a area utilization comparable to ours, achieving approximately 20% lower LUT usage but almost three times the BRAM usage. Additionally, their design's performance is more than twice as slow as ours.

**Table 4 Comparison of Dilithium-5 hardware implementations.**

| Ref | Device | Area (LUT/FF/DSP/BRAM) | Frequency [MHz] | Performance [$\mu s$] |
|---|---|---|---|---|
| **TW**[*] | ZUS+ | 17.1k/6.6k/4/12.5 | 375 | 64.7/131.0/72.0 |
| *Aikata et al. (2023a)*[*] | ZUS+ | 23.3k/9.8k/4/24 | 270 | 147.2/258.3/172.9 |
| *Aikata et al. (2023b)*[*] | ZUS+ | 18.5k/9.3k/4/24 | 200 | 194.2/342.1/228.9 |
| *Beckwith, Nguyen & Gaj (2021)* | VUS+ | 53.9k/28.4k/16/29 | 256 | 54.8/91.2/53.3 |
| *Li et al. (2024)* | ZUS+ | 32.0k/9.7k/24/14 | 300 | 36.3/184[1]/42.3 |
| *Gupta et al. (2023)* | ZUS+ | 14.0k/6.8k/4/35 | 391 | 161.0/291.0/173 |
| **TW**[*] | Artix-7 | 17.1k/6.6k/4/12.5 | 160 | 151.6/307.0/168.8 |
| *Zhao et al. (2021)* | Artix-7 | 30.0k/10.4k/10/11 | 96.9 | 90.5/163.0/93.3 |
| *Wang et al. (2022)* | Z-7000 | 21.0k/9.7k/10/28 | 159 | 127.0/592.8[1]/99.8 |

**Notes:**
[*] Unified design.
[1] Reports average results.

**Table 5 Comparison of Kyber-1024 hardware implementations.**

| Ref | Device | Area (LUT/FF/DSP/BRAM) | Frequency [MHz] | Performance [$\mu s$] |
|---|---|---|---|---|
| **TW**[*] | ZUS+ | 17.1k/6.6k/4/12.5 | 375 | 13.9/15.5/20.4 |
| *Aikata et al. (2023a)*[*] | ZUS+ | 23.3k/9.8k/4/24 | 270 | 33.7/42.0/51.5 |
| *Dang, Mohajerani & Gaj (2023)* | ZUS+ | 11.6k/11.6k/8/10.5 | 450 | 8.0/10.6/13.2 |
| *Ni et al. (2023)*[1] | ZUS+ | 17.8k/14.0k/2/0 | 435 | 6.2/7.8/9.4 |
| **TW**[*] | Artix-7 | 17.1k/6.6k/4/12.5 | 160 | 32.6/36.3/47.8 |
| *Xing & Li (2021)* | Artix-7 | 7.4k/4.6k/2/3 | 161 | 58.2/67.9/86.2 |
| *Bisheh-Niasar, Azarderakhsh & Mozaffari-Kermani (2021)* | Artix-7 | 10.5k/9.9k/8/13 | 200 | 17.3/20.6/31.3 |
| *Guo, Li & Kong (2022)* | Artix-7 | 7.9k/3.9k/4/16 | 159 | 49.1/52.8/66.0 |
| *Nguyen et al. (2024)* | Artix-7 | 5.5k/3.4k/2/3.5 | 185 | 45.9/54.6/69.7 |

**Notes:**
[*] Unified design.
[1] Results only for server-side.

Focusing on compactness, *Zhao et al. (2021)* and *Wang et al. (2022)* focus on reduced area usage while maintaining reasonable performance. *Zhao et al. (2021)* employ a segmented pipelining technique that reduces storage requirements and processing time. Although their BRAM usage is slightly lower than ours, their overall area utilization remains higher, while offering better performance. *Wang et al. (2022)*. implement only the core functions in hardware, relying on software for pre-processing and post-processing. Their design achieves minimal area usage, yet remains marginally higher than ours.

### Comparison with hardware implementations of Kyber

In Table 5, we detail the comparison with standalone Kyber hardware implementations. In particular, each work presents results for different security levels, and some use distinct designs for client-and server-side computations, with the exception of *Nguyen et al. (2024)*, who report results across all security levels.

Among high-performance designs, *Dang, Mohajerani & Gaj (2023)* achieve low latency through k-polynomials parallel execution. Their design has similar area utilization: they

use fewer LUTs and BRAMs, whereas our design uses fewer FFs and DSPs, while achieving slightly better performance. An even more performant design by *Ni et al. (2023)* leverages efficient pipelining and FIFO-based buffering for maximum parallelization. While *Ni et al. (2023)*'s architecture uses zero BRAMs, their FF utilization is notably higher. *Bisheh-Niasar, Azarderakhsh & Mozaffari-Kermani (2021)* aimed at high-performance on more constrained FPGA. Their optimized design achieves the best frequency on the Artix-7 platform.

Other studies, such as those by *Xing & Li (2021)*, *Guo, Li & Kong (2022)* and *Nguyen et al. (2024)*, focus on compact implementations while maintaining a solid performance. Each of these designs utilizes less hardware resources, which is expected, as Dilithium in our combined design brings higher area utilization. In terms of performance, our design significantly outperforms these with about 40% reduction in execution time.

Overall, these results indicate that standalone Kyber implementations, optimized for efficiency, would be better suited for applications that do not require a digital signature algorithm.

## DISCUSSION

The results in the previous section demonstrate significant advantages of the unified design, particularly with regard to reduced area utilization. This reduction stems from resource sharing between the DSA and KEM units, highlighting a key efficiency: In a scenario where a standalone implementation is used for DSA and KEM separately, each would require unique resources. In our unified design, however, the implementation of Kyber requires minimal additional area, as most resources are already allocated for Dilithium. This essentially renders Kyber support nearly "free" in terms of area costs within the unified framework, a substantial advantage for system design. This unified approach is beneficial in two primary contexts. First, on high-efficiency server platforms, where support for multiple cryptographic schemes is necessary, the unified design provides streamlined resource use and flexibility.

### Practical deployment

Server environments often require varied cryptographic algorithms, and this design supports multiple variants without having to replicate resource blocks for each scheme independently. Especially servers and gateways in high-speed communication networks can deploy this efficient solution to manage secure sessions. Second, for resource-constrained platforms, such as IoT devices, the unified design offers an efficient means to implement both DSA and KEM without the need for separate hardware for each function. This approach not only reduces the area, but may also lead to lower power consumption, which is crucial for battery-operated or low-power devices.

### Implementation attacks

While the theoretical security of both cryptographic schemes is well-established, implementation-specific vulnerabilities remain a concern. Potential threats include side-channel attacks, as demonstrated in works such as *Primas, Pessl & Mangard (2017)*,

*Kim et al. (2020)*, *Zhao et al. (2023)*, and fault-injection attacks, such as *Bindel, Buchmann & Krämer (2016)*, *Ravi et al. (2023)*. Hardware implementations generally offer enhanced resistance to such attacks due to reduced signal-to-noise ratios and parallel processing capabilities. However, these features alone may not fully prevent information leakage through power consumption or electromagnetic emissions, which could reveal sensitive data.

To mitigate basic side-channel attacks, such as those exploiting timing execution leakage, we ensured that all secret-sensitive operations in our design are implemented in constant time, consistent with reference implementations. Against more sophisticated attacks, including simple or differential power analysis, as well as template-or deep learning-based attacks, countermeasures such as masking or shuffling could be integrated.

These advanced countermeasures will be explored in future work, where we plan to conduct a detailed evaluation of their resource utilization and potential impact on performance.

### Open problems

Future work could explore further optimizations especially for constrained hardware platforms, small FPGA boards. One such modification could involve reducing the number of coefficients processed in parallel to further minimize area usage. Another possible optimization is to tailor the design to specific variants of the schemes, rather than supporting all variants universally. These changes would enable a more application-specific implementation that conserves both area and power.

Another open issue that remains is a deep investigation of side-channel leakage, which is possible despite the designed solution deploying parallel processing, reducing the signal-to-noise ratio, thus preventing simple tracing of secret data.

## CONCLUSION

In this article, we proposed a set of novel optimization techniques for the unified hardware implementation of two leading post-quantum cryptographic schemes: Dilithium and Kyber. The optimization steps mainly dealt with new improved memory management, operation schedule, and more efficient parts such as polynomial arithmetic unit, polynomial sample unit, and compression unit. Our design focuses on achieving a balance between area efficiency and high performance, with the top component utilizing only 17.1k LUTs, 6.6k FFs, four DSPs, and 12.5 BRAMs, while achieving a working frequency of 375 MHz on the high-efficiency Zynq Ultrascale + platform, respectively, working of 160 MHz on the resource-constrained Artix-7 platform. These figures represent a significant improvement over existing unified architectures, particularly with a nearly 50% reduction in BRAM usage compared to *Aikata et al. (2023a)*. Moreover, we demonstrate that our unified solution is even comparable with standalone implementations in terms of hardware resources and efficiency, but it saves costs by deploying only one compact implementation that can be beneficial in smaller and more cost-efficient FPGA platforms such as Artix-7.

As part of our future work, we plan to conduct a thorough side-channel leakage analysis to assess the vulnerability of our unified architecture. We will also explore and implement potential countermeasures aimed at mitigating these side-channel threats, with a particular focus on how these protections can be efficiently integrated within the unified design, ensuring both robust security and continued performance efficiency.

## ACKNOWLEDGEMENTS

During the preparation of this work, the authors used Chat GPT and Writefull to summarize information and paraphrase it for better presentation and understanding. After using this tool/service, the authors reviewed and edited the content as needed and take full responsibility for the content of the published article.

### Funding

This work is supported by the Ministry of the Interior of the Czech Republic under Grant VJ01010008. The funders had no role in study design, data collection and analysis, decision to publish, or preparation of the manuscript.

### Grant Disclosures

The following grant information was disclosed by the authors:
Ministry of the Interior of the Czech Republic: VJ01010008.

### Competing Interests

The authors declare that they have no competing interests.

### Author Contributions

- Patrik Dobias conceived and designed the experiments, performed the experiments, analyzed the data, performed the computation work, prepared figures and/or tables, authored or reviewed drafts of the article, and approved the final draft.
- Lukas Malina conceived and designed the experiments, analyzed the data, prepared figures and/or tables, authored or reviewed drafts of the article, and approved the final draft.
- Jan Hajny conceived and designed the experiments, analyzed the data, prepared figures and/or tables, authored or reviewed drafts of the article, and approved the final draft.

### Data Availability

The VHDL sources with Python testbenches are available at GitLab and Zenodo:

- https://gitlab.com/brno-axe/pqc/diky.

- Dobias, P., Malina, L., & Hajny, J. (2025). Efficient Unified Architecture for Post-Quantum Cryptography: Combining Dilithium and Kyber. Zenodo. https://doi.org/10.5281/zenodo.14891518.

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
