# Peer review of "Efficient unified architecture for post-quantum cryptography: combining Dilithium and Kyber"

_PeerJ Computer Science, doi:10.7717/peerj-cs.2746_

## Round 0.1 · original submission · Major Revisions

The referral process is now complete. While finding your paper interesting and worthy of publication, the referees and I feel that more work could be done before the paper is published. My decision is therefore to provisionally accept your paper subject to major revisions. More details are needed. The state of the art should be discussed in detail. Comparisons should be detailed. Implementation attacks should be discussed.

·

Basic reporting

The manuscript is written in clear and professional English, ensuring ease of understanding. It provides a well-defined context and background for the study, effectively highlighting the importance of post-quantum cryptography.

Experimental design

The study is methodically designed to address the research questions, with a rigorous experimental setup using FPGA platforms.

Methods are sufficiently detailed to replicate the experiments, particularly in the areas of hardware optimization and resource usage.

But may be some suggestion to improve are as follows:

The design focuses on high-efficiency platforms (e.g., Zynq Ultrascale+) but does not adequately address scalability to even more constrained environments.

While the manuscript briefly mentions side-channel attacks, a deeper exploration of security implications in the proposed architecture is necessary.

Validity of the findings

The validity of the findings in real-world applications could be strengthened by testing on a wider variety of FPGA platforms or through practical deployment scenarios.

The robustness of the proposed architecture in the face of potential implementation-specific attacks (e.g., side-channel or fault injection) has not been fully validated.

Reviewer 2 ·

Basic reporting

no comment

Experimental design

no comment

Validity of the findings

no comment

Additional comments

Overview
This article explores the potential of integrating two post-quantum schemes selected by NIST—the Dilithium digital signature scheme and the Kyber key encapsulation mechanism—into a unified architecture. It proposes a new optimization technique that achieves a balance between region efficiency and high performance. Compared to previous designs, this scheme demonstrates superior resource efficiency and performance, making it suitable for secure sessions in high-speed communication networks. Additionally, it has lower hardware resource requirements on small FPGA platforms, making it ideal for security applications in embedded devices and the Internet of Things.
Strengths
1.Innovativeness: A unified architecture for the most compact and high-performance hardware implementation of Diemithium and Kyber, supporting all security levels, has been proposed.
2. Practicality: Efficient solutions can also be deployed on small, low-cost, and low-density FPGA platforms.
3. Comprehensiveness: This article proposes a new optimization technique for the unified hardware implementation of two leading post-quantum cryptographic schemes: Dilithium and Kyber. A comparison is made among the unified hardware implementation, the Dilithium-5 hardware implementation, and the Kyber-1024 hardware implementation in terms of area, frequency, and performance.

Areas for Improvement
1. Content Depth: This solution achieves optimization in performance in Dilithium polynomial operations through the utilization of dual-port RAM and a specialized memory hierarchy. Nevertheless, the design of dual-port RAM and the multi-layered storage structure might render the hardware design and debugging more complicated, and also pose challenges in terms of memory utilization and design complexity.
2. Structural Clarity: Some paragraphs lack clear logical structure, making the argumentation less coherent. Reorganizing the content could help in presenting the points more clearly.
3. Lack of Examples: This article shows two examples of efficient scheduling, but does not give a legend explanation of the scheme in this article.

The English writing should be improved. The grammar mistakes, spelling mistakes, and improper descriptions should be taken more consideration. Some examples are given below:
a) Section Ⅰ, "the use of Dilithium's Number Theoretic Transform (NTT) multiplier for Saber multiplication as well" - here the "as The "well" may be somewhat confusing, seeming to imply that Dilithium's NTT multiplier is used directly for all of Saber's multiplications. This formulation may need clarification, especially in terms of how Saber's multiplication is reinforced.
b) Section Ⅱ, "first is employed during matrix A generation" : This section can be changed to "first is employed during the generation of matrix A "for more clarity.
c) Section Ⅱ, "the second occurs during the generation of secret key and error vectors" : It is suggested to change "the generation of secret key and error vectors" to "the generation of the secret key and the error vectors" for consisatency.
d) Section Ⅲ, “The selected depth of the temporary memory ensures that a sufficient number of polynomials are available in the memory...” the "are" might be a bit out of place here; consider replacing it with "is," because "the number" is singular.
e) Section Ⅲ, Sentence "Note that no additional memory is needed for this, as the memory requirements for Dilithium are substantially larger, as discussed in the previous section. "may leave the reader wondering why additional memory is not needed, more background information is suggested to make it clearer.

Reviewer 3 ·

Basic reporting

The author primarily researched the integrated hardware architecture of the post-quantum cryptographic digital signature scheme Dilithium and the KEM scheme Kyber. In the paper, the author proposed optimizations in modular multipliers, data flow, and scheduling. Finally, the design's implementation results and comparisons on FPGA were presented.

The article is written in fluent and clear language, making it easy to understand. The research terminology is used accurately, and the study includes a thorough literature review of the research background. The detailed description of the specific components of both algorithms is commendable.

Experimental design

The author claims to have proposed various optimization techniques and specific implementation methods to support the hardware design. However, I believe there are still several issues that need to be clarified:

1. Dual-port RAM is a common design in Kyber and Dilithium hardware implementations. The memory bit-width and data computation width should align accordingly. The author should specify the parallelism of the polynomial computation units corresponding to the bandwidth and how it meets the design requirements.

2. The butterfly unit is the fundamental computation structure in NTT. The structure claimed by the author employs extensive pipelining but shows no significant difference from conventional butterfly units in its specific implementation. The author should clarify how this structure ensures compatibility with both Kyber and Dilithium simultaneously.

3. The polynomial sampling unit is equally important in the overall hardware implementation. However, the author only briefly introduces the basic sampling process without presenting the related hardware implementation results, such as the hardware designs of Keccak and SHA-3, as well as the different sampling structures for Kyber and Dilithium.

4. The overlapping of multi-sampling and NTT computation is a commonly used technique in Kyber and Dilithium hardware designs. The author should elaborate on the advantages of the proposed design compared to prior designs in the literature.

Validity of the findings

In the implementation results and comparisons, the author used two different FPGAs for diverse data comparisons and drew some conclusions. However, I still have questions that need clarification:

1. In Figure 5, the author indicates that areas in different colors represent different components, and the compress component occupies a significant portion of the area. However, the compression process in Kyber only constitutes a small part, with low hardware resource requirements. The author should provide a reasonable explanation for this discrepancy.

2. The author's design demonstrates certain advantages in frequency, showing improved frequency performance compared to other designs. The author should specify which key optimizations and techniques contributed to this improvement in frequency performance.

---

## Round 0.2 · accepted · Accept

We are happy to inform you that your manuscript has been accepted for publication since the reviewers' comments have been addressed.

Reviewer 2 ·

Basic reporting

no

Experimental design

no

Validity of the findings

no

Additional comments

no

Reviewer 3 ·

Basic reporting

The authors have made reasonable modifications and edits based on the previous revision suggestions, effectively improving the structure, content presentation, and experimental data representation of the paper. Additionally, significant refinements have been made to the language, making the writing more fluent, clear, and aligned with academic writing standards. These enhancements further improve the readability and rigor of the paper, making it more valuable and relevant in the field of post-quantum cryptographic hardware implementations.

Experimental design

no comment

Validity of the findings

no comment

Additional comments

no comment